# Clinical Characteristics and Outcome of Hospitalized COVID-19 Patients Treated with Standard Dose of Dexamethasone or High Dose of Methylprednisolone

**DOI:** 10.3390/biomedicines10071548

**Published:** 2022-06-29

**Authors:** Alessandro Russo, Chiara Davoli, Cristian Borrazzo, Vincenzo Olivadese, Giancarlo Ceccarelli, Paolo Fusco, Alessandro Lazzaro, Rosaria Lionello, Marco Ricchio, Francesca Serapide, Bruno Tassone, Elio Gentilini Cacciola, Claudio Maria Mastroianni, Carlo Torti, Gabriella d’Ettorre, Enrico Maria Trecarichi

**Affiliations:** 1Infectious and Tropical Disease Unit, Department of Medical and Surgical Sciences, “Magna Graecia” University of Catanzaro, 88100 Catanzaro, Italy; chiara.davoli93@gmail.com (C.D.); olivadesevincenzo.95@gmail.com (V.O.); paolofusco89@gmail.com (P.F.); rosarialionello0@gmail.com (R.L.); marco.ricchio@libero.it (M.R.); francescaserapide@gmail.com (F.S.); tassonebruno89@gmail.com (B.T.); torti@unicz.it (C.T.); em.trecarichi@unicz.it (E.M.T.); 2Infectious and Tropical Disease Unit, Department of Public Health and Infectious Diseases, “Sapienza” University of Rome, 00185 Roma, Italy; cristian.borrazzo@uniroma1.it (C.B.); giancarlo.ceccarelli@uniroma1.it (G.C.); alessandro1lazzaro@gmail.com (A.L.); gentilini.1979701@studenti.uniroma1.it (E.G.C.); claudio.mastroianni@uniroma1.it (C.M.M.); gabriella.dettorre@uniroma1.it (G.d.)

**Keywords:** SARS-CoV-2, COVID-19, dexamethasone, methylprednisolone, mortality

## Abstract

The hyperinflammatory phase represents the main cause for the clinical worsening of acute respiratory distress syndrome (ARDS) in Coronavirus disease 2019 (COVID-19), leading to the hypothesis that steroid therapy could be a mainstream treatment in COVID-19 patients. This is an observational study including all consecutive patients admitted to two Italian University Hospitals for COVID-19 from March 2020 to December 2021. The aim of this study was to describe clinical characteristics and outcome parameters of hospitalized COVID-19 patients treated with dexamethasone 6 mg once daily (standard-dose group) or methylprednisolone 40 mg twice daily (high-dose group). The primary outcome was the impact of these different steroid treatments on 30-day mortality. During the study period, 990 patients were evaluated: 695 (70.2%) receiving standard dosage of dexamethasone and 295 (29.8%) receiving a high dose of methylprednisolone. Cox regression analysis showed that chronic obstructive pulmonary disease (HR 1.98, CI95% 1.34–9.81, *p* = 0.002), chronic kidney disease (HR 5.21, CI95% 1.48–22.23, *p* = 0.001), oncologic disease (HR 2.81, CI95% 1.45–19.8, *p* = 0.005) and high-flow nasal cannula, continuous positive airway pressure or non-invasive ventilation oxygen therapy (HR 61.1, CI95% 5.12–511.1, *p* < 0.001) were independently associated with 30-day mortality; conversely, high-dose steroid therapy was associated with survival (HR 0.42, CI95% 0.38–0.86, *p* = 0.002) at 30 days. Kaplan–Meier curves for 30-day survival displayed a statistically significant better survival rate in patients treated with high-dose steroid therapy (*p* = 0.018). The results of this study highlighted that the use of high-dose methylprednisolone, compared to dexamethasone 6 mg once daily, in hospitalized patients with COVID-19 may be associated with a significant reduction in mortality.

## 1. Introduction

SARS-CoV-2 infection can progress to a severe respiratory illness that requires hospital care. In a significant percentage of cases, it can progress to a critical disease, ranging from mild respiratory failure to severe acute respiratory distress syndrome (ARDS).

The hyperinflammatory phase, in which an excessive and ineffective immune system activation leads to a massive infiltration of immune cells in the lungs and a systemic overproduction of pro-inflammatory cytokines, represents the main cause for ARDS [1,2,3] leading to the hypothesis that steroid therapy could be a mainstream treatment for Coronavirus disease 2019 (COVID-19).

The RECOVERY trial [4] demonstrated for the first time that steroids (dexamethasone in particular) are a standard of care treatment for patients with COVID-19 necessitating oxygen supplementation. However, it is not clear whether a dose of steroids greater than that used in the RECOVERY trial could lead to better outcomes in patients with more severe forms of COVID-19 [5,6,7].

The aim of this study was to describe clinical characteristics and outcome parameters of hospitalized COVID-19 patients treated with a standard steroid dose (dexamethasone 6 mg once daily) or a higher dose of steroids (methylprednisolone 40 mg twice daily). Since the severity of the disease could impact the actual benefit of increasing steroid dose and could, therefore, act as a major confounder, we stratified patients treated with low-flow oxygen therapy and those treated with high-flow nasal cannula, continuous positive airway pressure or non-invasive ventilation (HFNC/CPAP/NIV); furthermore, possible risk factors for 30-day mortality were investigated.

## 2. Materials and Methods

### 2.1. Study Design and Data Collection

This observational study included all consecutive patients admitted to two Italian University Hospitals: the “Mater Domini” teaching hospital of Catanzaro (Calabria, Southern Italy) and the “Umberto I” teaching hospital of Rome (Lazio, Central Italy), from March 2020 to December 2021.

The Ethics Committees of each participating center approved the study protocol for observational analyses. For the “Umberto I” teaching hospital of Rome, the study was designed as a prospective, and was approved by the local Ethics Committee (n. 5923/2020) with informed consent obtained from each patient included. For the “Mater Domini” teaching hospital of Catanzaro, this study was designed as a retrospective; it was notified to the Ethics Committee of the Calabria Region on 13 May 2020. The study did not require ethical approval by an Ethics Committee as determined by the Italian Drug Agency, note 20 March 2008 (GU Serie Generale no. 76 31 March 2008). The need for written informed consent was waived for patients owing to the retrospective nature of the study. The study was conducted in accordance with the Declaration of Helsinki and the Good Clinical Practice standard.

Inclusion criteria were: (1) a positive SARS-CoV-2 real-time polymerase chain reaction test on nasopharyngeal swab, according to WHO recommendations [8]; (2) the need for hospitalization for COVID-19; (3) treatment with steroid therapy.

The following data were extracted from the patients’ medical databases of each participating center: demographics, clinical and laboratory findings, comorbidities, microbiologic data, COVID-19 diagnosis date, radiological characteristics of pneumonia, therapies administered, need for oxygen or ventilation support during the hospital stay, length of ICU stay, length of hospital stay and outcome. All patients were followed until discharge or death. Non-invasive respiratory support techniques included HFNC, CPAP and NIV. Management of patients at admission and during hospitalization were previously reported [9].

### 2.2. Steroid Dosage

Dexamethasone 6 mg once daily (standard-dose therapy) or methylprednisolone 40 mg twice daily (high-dose therapy) were used to treat patients based on clinical characteristics and oxygen support, upon the attending physician’s clinical judgement and outside pre-defined protocols [10]. Steroids were administered for at least 10 days, or less in cases of discharge or death. Patients who received a standard dexamethasone dose were evaluated for a switch to a high dose of methylprednisolone (40 mg twice daily) if clinical worsening: these patients were excluded from the final analysis.

### 2.3. Outcome and Statistical Analysis

The primary outcome was 30-day mortality in hospitalized patients with COVID-19. The Welch’s *t*-tests assuming unequal variances were used for continuous independent variables, while Pearson’s chi-square or Fisher’s exact test were used for categorical variables when appropriate. The ANOVA test (Welch’s analysis of variance) was used to assess group differences for continuous outcomes. Welch’s *t*-tests assuming unequal variances were used for post-hoc comparisons. Results were expressed as the mean and/or median, when appropriate, with standard deviation (±SD) and/or interquartile range (IQR, 25–75%) for non-continuous normally distributed variables (Shapiro–Wilk test) and as a sample (*n*) and percentage (%) for categorical variables. All tests were two-tailed, and a *p*-value less than 0.05 was considered statistically significant. Multivariate analysis was used to identify independent predictors of 30-day mortality. Matched bivariate analyses were conducted using a conditional logistic regression model with a stepwise method, incorporating all variables found to be significant at the univariate analysis (*p* < 0.05). A matched multivariate model was constructed using Cox proportional hazards regression when appropriate, accounting for clustering on matched pairs (with standard steroid dose and those of patients treated with a higher dose of steroid). The final selected model was tested for possible confounding. In addition, 95% confidence intervals were calculated for hazard ratio. Survival was analyzed by Kaplan–Meier curves. All data were analyzed using a commercially available statistical software package (SPSS Statistics for Mac, 25.0; IBM Corp., Armonk, NY, USA).

## 3. Results

During the study period, a total of 1054 COVID-19 patients was enrolled. Out of these, 64 patients did not receive any steroid therapy, therefore, according to the inclusion criteria, they were excluded from the analysis. Overall, 990 patients were evaluated: 695 (70.2%) receiving a standard dosage of dexamethasone (6 mg once daily) and 295 (29.8%) receiving a high dose of methylprednisolone (40 mg twice daily).

Table 1 reports the baseline clinical features, demographic information, laboratory values, pharmacological characteristics and outcome of the study population. Median age was 69.3 years (IQR, 58–81) and 45.1% were males. The most frequent comorbidities were cardiovascular disease other than arterial hypertension (34.3%) and arterial hypertension (20.7%), chronic obstructive pulmonary disease (21.3%), chronic kidney disease (15.2%), diabetes mellitus type 2 (12.6%) and obesity (10.9%). The most frequent clinical features at the time of hospitalization were fever (59.1%), dyspnoea (44.7%), cough (40.5%), asthenia (34%) and diarrhoea (11.1%). Overall, 671 (67.7%) patients were treated with low-flow oxygen, while 319 (32.3%) received HFNC/CPAP/NIV support. Regarding antibiotics, the most frequently used were: cephalosporins (32%), macrolides (30%), carbapenems (21%). Overall, 198 patients (20%) died at 30 days.

As shown in Table 2, compared to patients who received a high dose, those who received a standard dose of steroids were more likely to be older (71.7 [IQR, 58.5–82.8] vs. 66.3 [IQR 55.3 vs. 80] years, *p* = 0.001) and male (52.3% vs. 28.1%, *p* < 0.001), suffered more frequently from a cardiovascular disease other than arterial hypertension (41% vs. 18.6%, *p* < 0.001), chronic kidney disease (18.2% vs. 8.1%, *p* < 0.001) and/or chronic obstructive pulmonary disease (25.6% vs. 11.1%, *p* < 0.001), reported asthenia (38.8% vs. 22.7%, *p* < 0.001), had higher PaO_2_/FiO_2_ ratio (330 [IQR, 229–386] vs. 286 [IQR, 172–357], *p* < 0.001) and C-reactive protein levels (11.9 [IQR, 6.7–21.9] vs. 7.9 [IQR, 2.3–37.5] mg/L, *p* = 0.009); by contrast, they were less likely to suffer from diabetes mellitus (9.6% vs. 19.6%, *p* < 0.001) and to display dyspnoea (41% vs. 53.5%, *p* < 0.001).

Regarding pharmacological and oxygen therapies and outcomes (Table 3), remdesivir (15.2% vs. 55.5%, *p* < 0.001), low-weight molecular heparin (76.9% vs. 36.2%, *p* < 0.001), antibiotics (78.3% vs. 65%, *p* = 0.001) and low-flow oxygen (81.6% vs. 61.8%, *p* < 0.001) were more frequently administered to patients in the high-dose steroid group, compared to the standard-dose group. The overall 30-day mortality rate was significantly higher for patients in the standard-dose group (166/695, 23.8%) than for those in the high-dose group (32/295, 10.8%; *p* < 0.001).

Table 4 shows the comparison between survivors and non-survivors. Non-survivors were more frequently older (82 [IQR, 73–88] vs. 65 [IQR, 56–80] years, *p* < 0.001), suffered from cardiovascular disease other than arterial hypertension (66.1% vs. 26.4%, *p* < 0.001), chronic kidney disease (25.7% vs 12.6%, *p* = 0.002) and chronic obstructive pulmonary disease (56% vs. 12.6%, *p* < 0.001); they had a higher respiratory rate (27 (IQR, 20–30) vs. 20 [IQR, 17–24] acts/min, *p* = 0.009) and lower PaO_2_/FiO_2_ ratio (310 [IQR, 142.8–362.5] vs. 316 [IQR, 228.8–376.4], *p* = 0.001). In addition, compared to survivors, non-survivors had higher levels of interleukin-6 (49.8 [IQR, 24.6–112.6] vs. 18.9 [IQR, 8.3–43] pg/mL, *p* = 0.032) and D-dimer (0.9 [IQR, 0.4–1.5] vs. 0.7 [IQR, 0.4–1.3] mg/L, *p* = 0.043); moreover, they were less frequently treated with remdesivir (10.1% vs. 31.5%, *p* < 0.001), low-weight molecular heparin (29.8% vs. 53%, *p* < 0.001), high-dose steroid therapy (16.6% vs. 33%, *p* < 0.001) and low-flow oxygen (44.4% vs. 73.6%, *p* < 0.001); instead, non-survivors more frequently received HFNC/CPAP/NIV (70.2% vs. 22.7%, *p* < 0.001) (see Table 5).

Cox regression analysis (reported in Table 6) showed that risk factors independently associated with 30-day mortality were chronic obstructive pulmonary disease (HR 1.98, CI95% 1.34–9.81, *p* = 0.002), chronic kidney disease (HR 5.21, CI95% 1.48–22.23, *p* = 0.001), oncologic disease (HR 2.81, CI95% 1.45–19.8, *p* = 0.005) and HFNC/CPAP/NIV oxygen therapy (HR 61.1, CI95% 5.12–511.1, *p* < 0.001); conversely, high-dose steroid therapy was associated with survival (HR 0.42, CI95% 0.38–0.86, *p* = 0.002).

Figure 1 shows the Kaplan–Meier curves for 30-day survival in the COVID-19 population treated with standard-dose or high-dose steroids, displaying a statistically significant better survival in patients treated with high-dose steroid therapy (*p* = 0.018).

Kaplan–Meier curves in COVID-19 patients treated with high-dose steroid therapy or standard-dose steroid therapy in the subgroups of those treated with HFNC/CPAP/NIV or low-flow oxygen are reported in Figure 2. No statistically significant differences were observed in terms of survival in the 2 subgroups treated with low-flow oxygen (*p* = 0.457) or with HFNC/CPAP/NIV (*p* = 0.842).

## 4. Discussion

This observational real-life study primarily aimed to investigate the clinical impact of the use of two different dosages of steroids on 30-day mortality in a large population of patients hospitalized for COVID-19.

In our large cohort, treatment with a high-dose corticosteroid (i.e., methylprednisolone 40 mg twice daily) was significantly associated with a lower 30-day mortality compared to the outcome of treatment with a standard dose (i.e., dexamethasone 6 mg once daily) in Cox regression analysis and survival analysis by using Kaplan–Meier curves. However, when the impact on mortality of the two steroid regimens was analyzed in the subgroups of patients treated or not with HFNC/CPAP/NIV, no statistically significant differences were found.

The strength of our study is that the effect of high-dose methylprednisolone on survival was significantly higher on the overall population if compared to dexamethasone, and it is possible to hypothesize that methylprednisolone could be at least as effective as dexamethasone in the treatment of COVID-19. This could be also explained by a better penetration of methylprednisolone in the lung tissue, in comparison to dexamethasone [11,12,13,14], although no data are available in severe COVID-19 patients. Of importance, none of the previous studies analyzed the impact of methylprednisolone vs. dexamethasone in subgroups of patients treated with a high vs. low flow of oxygen.

There are some important limitations to our study. First, the retrospective nature of this study involving two heterogeneous populations belonging to different centers; moreover, data are refereed to different waves of the pandemic and therapeutic approach was consistently modified during these years; third, microbiological endpoints, such as time to negative nasal swab or bacterial or fungal co- and super-infections, were not considered in the final analysis; finally, in this cohort, consecutively hospitalized patients were enrolled independently from COVID-19 severity and no data are available about follow-up after discharge.

During the first month of the pandemic, several authors recommended prescribing steroids based on anecdotal observations and retrospective uncontrolled series of patients [15,16,17,18]. In contrast, other studies argued that corticosteroids may be harmful and cause delayed viral clearance in COVID-19 patients [19], as previously reported also in SARS patients [20]. As reported below, several studies and randomized clinical trials tried to inquire into steroid therapy efficacy with the aim to define the most appropriate drug, dose and duration of steroid therapy that should be administered to improve the clinical outcomes of COVID-19.

In July 2020, the RECOVERY Collaborative Group showed in a preliminary report, and confirmed in the definitive publication in February 2021, that in patients with COVID-19, therapy with 6 mg once daily for up to 10 days (or until hospital discharge if sooner) of oral or intravenous dexamethasone resulted in significantly lower 28-day mortality among those undergoing oxygen supplementation, but not among those receiving no respiratory support [4]. Current recommendations by international guidelines on the use of steroids in COVID-19 patients are based on the findings of this trial [21,22].

In September 2020, the WHO Rapid Evidence Appraisal for COVID-19 Therapies (REACT) Working Group published the results of a prospective meta-analysis that pooled data from seven randomized clinical trials (including the RECOVERY trial) that evaluated the efficacy of corticosteroids in 1703 critically ill patients with COVID-19 [23]. They found that administration of systemic corticosteroids (dexamethasone, hydrocortisone and methylprednisolone), compared to usual care or placebo, was associated with a significantly lower 28-day all-cause mortality. After stratifying the studies according to corticosteroid drug, a significant association between steroid treatment and survival was demonstrated only for dexamethasone, and not for hydrocortisone and methylprednisolone. However, only in one of the seven clinical trials included (Steroids-SARI), methylprednisolone (at dosage of 40 mg twice daily for five days) was the drug investigated [24]; a total of 47 patients was enrolled in this trial, so that this study contributed to only 3.5% of the weight in the primary meta-analysis.

Further studies investigated the efficacy of treatment with methylprednisolone in COVID-19 patients [5,6,25,26]. In particular, the COVID-19-Metcovid trial evaluated the efficacy of methylprednisolone at a dosage of 0.5 mg/kg twice daily for 5 days compared to placebo in hospitalized patients with suspected COVID-19 requiring supplementary oxygen or mechanical ventilation [25]. In this trial, the authors did not find any significant difference for 28-day mortality between the groups in the overall population, but only in the subgroup of patients aged > 60 years. By contrast, in the trial conducted by Ranjbar et al. [5] comparing patients diagnosed with COVID-19 and respiratory impairment (i.e., O_2_ saturation of less than 92 in room air on admission) randomly assigned to receive a 10-day course of methylprednisolone (2 mg per kilogram intravenously) vs. dexamethasone (6 mg intravenously) with the standard care, the all-cause 28-day mortality rates were lower in patients treated with methylprednisolone than in those who received dexamethasone, though the difference did not reach statistical significance. Instead, they reported a significantly better clinical status at day 5 and day 10 of admission, lower mean length of hospital stay and need for a ventilator in the intervention group than in the control group [5].

A more recent prospective study published on February 2022 by Saeed et al. compared the differences in clinical outcome and laboratory results in mechanically ventilated patients with SARS-CoV-2 radiologically confirmed pneumonia treated for ten days with dexamethasone 6 mg/day (192 patients) vs. those treated with methylprednisolone 2 mg/kg/day [26]. They reported that patients treated with methylprednisolone showed a statistically significant improvement in the mechanical ventilation days and length of stay in the intensive care unit, together with the overall mortality and severity inflammatory markers of cytokine storm, compared to those treated with dexamethasone after 10 days. Similar results were reported in the observational study conducted by Pinzon et al. [6].

## 5. Conclusions

In conclusion, these real-life experience data highlight that the use of methylprednisolone in hospitalized patients with COVID-19 may be associated with a significant reduction in mortality in the global population, but this difference was not observed when stratifying patients undergoing HFNC/CPAP/NIV or not. These results suggest the need to conduct further RCTs to evaluate the impact of different dose and duration of corticosteroid therapy in hospitalized COVID-19 patients at different stages of the disease.

## Figures and Tables

**Figure 1 biomedicines-10-01548-f001:**
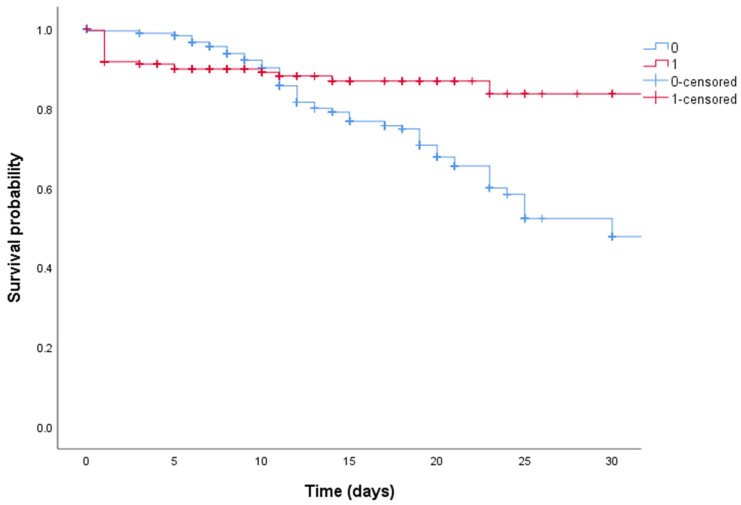
Kaplan–Meier curves of all COVID-19 patients treated with high-dose steroid therapy (red line) or standard-dose steroid therapy (blue line).

**Figure 2 biomedicines-10-01548-f002:**
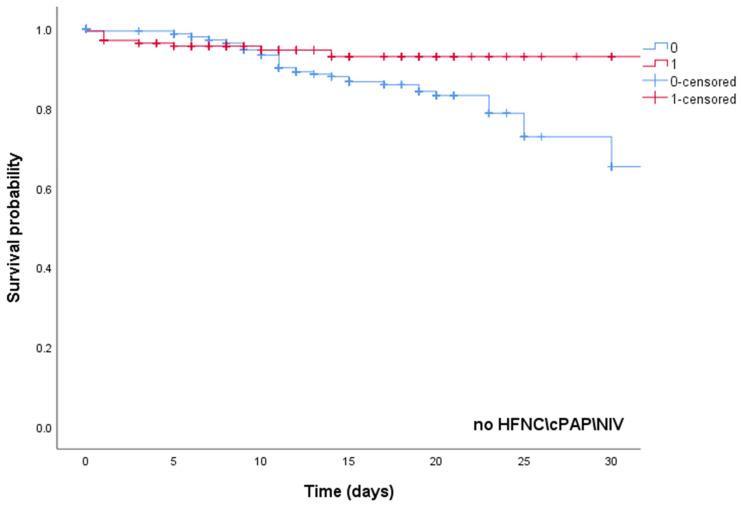
Kaplan–Meier curves of COVID-19 patients treated with high-dose steroid therapy (red line) or standard-dose steroid therapy (blue line) in the subgroups with no HFNC/CPAP/NIV use (*p* = 0.457) or with HFNC/CPAP/NIV use (*p* = 0.842). HFNC—high-flow nasal cannula; NIV—non-invasive ventilation; CPAP—continuous positive airway pressure.

**Table 1 biomedicines-10-01548-t001:** Baseline characteristics of study population.

Variables	All Patients (*n* = 990)
**Demographic information**	
Age (years), median [IQR 25–75%]	69.3 (58–81)
Male sex	447 (45.1)
**Comorbidities [*n* (%)]**	
Arterial hypertension	205 (20.7)
Cardiovascular disease other than arterial hypertension	340 (34.3)
Obesity	108 (10.9)
Diabetes mellitus type 2	125 (12.6)
Chronic kidney disease	151 (15.2)
Chronic obstructive pulmonary disease	211 (21.3)
Oncologic disease	87 (8.7)
Psychiatric disorders	48 (4.8)
Neurologic disorders	60 (6)
**Clinical features** **[*n* (%)]**	
Fever	586 (59.1)
Cough	401 (40.5)
Dyspnoea	443 (44.7)
Headache	16 (1.6)
Diarrhoea	111 (11.1)
Asthenia	337 (34)
Ageusia	77 (7.7)
Anosmia	89 (9)
**Lung function parameters [median (IQR 25–75%)]**	
Respiratory rate (acts/min)	21 (17–26)
PaO_2_/FiO_2_ ratio	314 (216–374)
**Laboratory values [median (IQR 25–75%)]**	
Interleukin-6 (pg/mL)	22 (9–45.3)
D-dimer (mg/L)	0.75 (0.4–1.3)
Fibrinogen (mg/dL)	520 (451–580)
Lactate dehydrogenase (UI/L)	357 (267–474)
Ferritin (ng/mL)	507 (207.3–867.5)
Platelets count (cells/µL)	213 (163–268.8)
Lymphocytes count (cells/µL)	930 (625–1400)
C-reactive protein (mg/L)	11 (4.8–27)
Procalcitonin (ng/mL)	0.08 (0.1–0.2)
**Pharmacological Therapy**	
Antibiotic	683 (69)
Standard-dose steroid	695 (70.2)
High-dose steroid	295 (29.8)
Remdesivir	270 (27.2)
Tocilizumab	17 (1.7)
Low-weight molecular heparin	479 (48.3)
**Oxygen therapy**	
Low-flow oxygen	671 (67.7)
HFNC/CPAP/NIV	319 (32.3)
**Outcomes**	
30-day mortality	198 (20)

HFNC—high-flow nasal cannula; NIV—non-invasive ventilation; CPAP—continuous positive airway pressure.

**Table 2 biomedicines-10-01548-t002:** Comparison between COVID-19 patients treated with standard- or high-dose steroids.

Variables	Standard-Dose Steroid (*n* = 695)	High-Dose Steroid (*n* = 295)	*p*-Value
**Demographic information**			
Age (years), median [IQR 25–75%]	71.7 (58.5–82.8)	66.3 (55.3–80)	**0.001**
Male sex	364 (52.3)	83 (28.1)	**<0.001**
**Comorbidities [*n* (%)]**			
Arterial hypertension	142 (20.4)	63 (21.3)	0.255
Cardiovascular disease other than Arterial hypertension	285 (41)	55 (18.6)	**<0.001**
Obesity	75 (10.8)	33 (11.1)	0.577
Diabetes mellitus type 2	67 (9.6)	58 (19.6)	**<0.001**
Chronic kidney disease	127 (18.2)	24 (8.1)	**<0.001**
Chronic obstructive pulmonary disease	178 (25.6)	33 (11.1)	**<0.001**
Oncologic disease	77 (11)	10 (3.3)	0.052
Psychiatric disorders	30 (4.3)	18 (6.1)	0.158
Neurologic disorders	42 (6)	18 (6.1)	0.953
**Clinical features** **[*n* (%)]**			
Fever	390 (56.1)	196 (66.4)	**0.001**
Cough	280 (40.2)	121 (41)	0.885
Dyspnoea	285 (41)	158 (53.5)	**<0.001**
Headache	15 (2.1)	1 (0.3)	0.122
Diarrhoea	76 (10.9)	35 (11.8)	0.822
Asthenia	270 (38.8)	67 (22.7)	**<0.001**
Ageusia	57 (8.2)	20 (6.7)	0.681
Anosmia	71 (10.2)	18 (6.1)	0.165
**Lung function parameters [median (IQR 25–75%)]**			
Respiratory rate (acts/min)	18 (16–24)	24 (19.5–28.5)	**<0.001**
PaO_2_/FiO_2_ ratio	320 (229–386)	286 (172–357)	**<0.001**
**Laboratory values [median (IQR 25–75%)]**			
Interleukin-6 (pg/mL)	17.6 (8.4–40.8)	35.3 (15.9–64.4)	0.41
D-dimer (mg/L)	0.7 (0.4–1.3)	0.71 (0.37–1.31)	0.914
Fibrinogen (mg/dL)	512 (436–574)	546 (467.8–614.5)	**0.005**
Lactate dehydrogenase (UI/L)	373 (278–473.5)	334 (239–476)	0.233
Ferritin (ng/mL)	492 (195–857)	569 (305–947)	**0.007**
Platelets count (cells/µL)	215 (164–268)	206 (159–270)	0.177
Lymphocytes count (cells/µL)	1000 (670–1490)	815 (552.5–1147.5)	0.078
C-reactive protein (mg/L)	11.9 (6.7–21.9)	7.9 (2.3–37.5)	**0.009**
Procalcitonin (ng/mL)	0.1 (0.1–0.2)	0.00 (0.00–0.10)	0.105

**Table 3 biomedicines-10-01548-t003:** Therapy and outcome parameters of COVID-19 patients treated with standard- or high-dose of steroids.

Variables	Standard-Dose Steroid (*n* = 695)	High-Dose Steroid (*n* = 295)	*p*-Value
**Pharmacological Therapy**			
Antibiotic	452 (65)	231 (78.3)	**0.001**
Remdesivir	106 (15.2)	164 (55.5)	**<0.001**
Tocilizumab	5 (0.7)	12 (4)	0.064
Low-weight molecular heparin	252 (36.2)	227 (76.9)	**<0.001**
**Oxygen therapy**			
Low-flow oxygen	430 (61.8)	241 (81.6)	**<0.001**
HFNC/CPAP/NIV	239 (34.3)	80 (27.1)	0.246
**Outcomes**			
30-day mortality	166 (23.8)	32 (10.8)	**<0.001**

HFNC—high-flow nasal cannula; NIV—non-invasive ventilation; CPAP—continuous positive airway pressure.

**Table 4 biomedicines-10-01548-t004:** Comparison between surviving and non-surviving COVID-19 patients.

Variables	Survivors (*n* = 792)	Non-Survivors (*n* = 198)	*p*-Value
**Demographic information**			
Age (years), median [IQR 25–75%]	65 (56–80)	82 (73–88)	**<0.001**
Male sex [*n* (%)]	347 (43.8)	100 (50.5)	0.381
**Comorbidities [*n* (%)]**			
Arterial hypertension	175 (22.1)	30 (15.1)	0.345
Cardiovascular disease other than Arterial hypertension	209 (26.4)	131 (66.1)	**<0.001**
Obesity	91 (11.4)	17 (8.6)	0.675
Diabetes mellitus type 2	88 (11.1)	37 (18.7)	0.355
Chronic kidney disease	100 (12.6)	51 (25.7)	**0.002**
Chronic obstructive pulmonary disease	100 (12.6)	111 (56)	**<0.001**
Oncologic disease	50 (6.3)	37 (18.7)	**0.002**
Psychiatric disorders	43 (5.4)	5 (2.5)	0.401
Neurologic disorders	50 (6.3)	10 (5)	0.891
**Clinical features [*n* (%)]**			
Fever	516 (65.1)	70 (35.3)	**<0.001**
Cough	361 (45.5)	40 (20.2)	**<0.001**
Dyspnoea	370 (46.7)	73 (36.8)	**0.03**
Headache	16 (2)	0	0.633
Diarrhoea	90 (11.3)	21 (10.6)	0.987
Asthenia	217 (27.4)	120 (60.6)	**<0.001**
Ageusia	66 (8.3)	11 (5.5)	0.482
Anosmia	74 (9.3)	15 (7.5)	0.633
**Lung function parameters [median (IQR 25–75%)]**			
Respiratory rate (acts/min)	20 (17–24)	27 (20–30)	**0.009**
PaO_2_/FiO_2_ ratio	316 (228.8–376.4)	310 (142.8–362.5)	**0.001**
**Laboratory values [median (IQR 25–75%)]**			
Interleukin-6 (pg/mL)	18.9 (8.3–43)	49.8 (24.6–112.6)	**0.032**
D-dimer (mg/L)	0.7 (0.4–1.3)	0.9 (0.4–1.5)	**0.043**
Fibrinogen (mg/dL)	517.5 (448.3–575)	560.5 (447.8–657.5)	0.089
Lactate dehydrogenase (UI/L)	336 (252–457)	457 (332–614.5)	**<0.001**
Ferritin (ng/mL)	506 (209–852)	551 (214–1078)	0.144
Platelets count (cells/µL)	210 (162–268)	226 (166.8–270.8)	0.147
Lymphocytes count (cells/µL)	950 (640–1380)	900 (600–1470)	0.323
C-reactive protein (mg/L)	10.5 (3.7–29)	13.1 (8.8–26.7)	0.160
Procalcitonin (ng/mL)	0.1 (0.0–0.2)	0.1 (0.1–0.6)	0.202

**Table 5 biomedicines-10-01548-t005:** Therapy and outcome of surviving or non-surviving COVID-19 patients.

Variables	Survivors (*n* = 792)	Non-Survivors (*n* = 198)	*p*-Value
**Pharmacological Therapy**			
Antibiotic	551 (69.5)	132 (66.6)	0.359
Remdesivir	250 (31.5)	20 (10.1)	**<0.001**
Tocilizumab	12 (1.5)	5 (2.5)	0.572
Low-weight molecular heparin	420 (53)	59 (29.8)	**<0.001**
Standard-dose steroid therapy	539 (68)	156 (78.8)	0.122
High-dose steroid therapy	262 (33)	33 (16.6)	**<0.001**
**Oxygen therapy**			
Low-flow oxygen	583 (73.6)	88 (44.4)	**<0.001**
HFNC/CPAP/NIV	180 (22.7)	139 (70.2)	**<0.001**

HFNC—high-flow nasal cannula; NIV—non-invasive ventilation; CPAP—continuous positive airway pressure.

**Table 6 biomedicines-10-01548-t006:** Cox regression analysis of risk factors associated with 30-day mortality in all study populations.

Variables	Hazard Ratio	CI95%-Lower	CI95%-Upper	*p*-Value
Chronic kidney disease	5.21	1.48	22.23	0.001
Oncologic disease	2.81	1.45	19.8	0.005
Chronic obstructive pulmonary disease	1.98	1.34	9.81	0.002
High-dose steroid therapy	0.42	0.38	0.86	0.002
HFNC/CPAP/NIV	61.1	5.12	511.1	<0.001

HFNC—high-flow nasal cannula; NIV—non-invasive ventilation; CPAP—continuous positive airway pressure.

## Data Availability

Data supporting reported results can be found at a.russo@unicz.it.

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
