# Peer review of "Clinical Characteristics and Outcome of Hospitalized COVID-19 Patients Treated with Standard Dose of Dexamethasone or High Dose of Methylprednisolone"

_biomedicines, 2022, doi:10.3390/biomedicines10071548_

Round 1

Reviewer 1 Report

Brief Summary

The authors conducted an observational study (retrospective data analysis as well as prospective data) with hospitalized COVID-19 patients that received standard or high-dose steroid treatment. They evaluated 990 patients and found a reduction of 30-day mortality in patients that received high-dose steroid treatment compared to those receiving the standard-dose. 

General comments

Overall, this is an interesting article. While reading I was a little distracted by the grammatical and spelling mistakes, the missing "s", too many commas, missing words, etc. I think the content is fine, but the writing/ the language should be improved. I did not list all of those errors in my specific comments - I would recommend to have the article proof-read by a native speaker. 

Specific comments

Line 13: Better use reason or cause instead of responsible

Line 16: including instead of included

Line 18 outcome parameters instead of outcome

Line 20: steroid treatments instead of steroid`s treatment

Line 30: Maybe "The results of this study highlighted..." instead of data highlighted

Line 49: The aim of this study...and outcome parameters....

Liene 52: ...could impact the actual benefit...and therefore acting as a major confounder...

Table 1: Instead of "All population" maybe a better phrase would be "all patients".

Table 1: In the line where the age is presented a bracket ist missing - I assume before IQR.

Table 2 Title: .either "with a standard- or high-dose of steroids" or "with standard- or high-dose steroids"

Table 2: I think there is again a bracket missing where the age is displayed.

Line 145: ... to the standrad-dose group"

Table 3: Maybe "outcome parameters"

Table 3: Why is the low flow oxygen and HFNC/etc in italic?

Table 4: Bracket missing within the age line.

Author Response

Brief Summary

The authors conducted an observational study (retrospective data analysis as well as prospective data) with hospitalized COVID-19 patients that received standard or high-dose steroid treatment. They evaluated 990 patients and found a reduction of 30-day mortality in patients that received high-dose steroid treatment compared to those receiving the standard-dose. 

General comments

Overall, this is an interesting article. While reading I was a little distracted by the grammatical and spelling mistakes, the missing "s", too many commas, missing words, etc. I think the content is fine, but the writing/ the language should be improved. I did not list all of those errors in my specific comments - I would recommend to have the article proof-read by a native speaker. 

Specific comments

Line 13: Better use reason or cause instead of responsible

Line 16: including instead of included

Line 18 outcome parameters instead of outcome

Line 20: steroid treatments instead of steroid`s treatment

Line 30: Maybe "The results of this study highlighted..." instead of data highlighted

Line 49: The aim of this study...and outcome parameters....

Liene 52: ...could impact the actual benefit...and therefore acting as a major confounder...

Table 1: Instead of "All population" maybe a better phrase would be "all patients".

Table 1: In the line where the age is presented a bracket ist missing - I assume before IQR.

Table 2 Title: .either "with a standard- or high-dose of steroids" or "with standard- or high-dose steroids"

Table 2: I think there is again a bracket missing where the age is displayed.

Line 145: ... to the standrad-dose group"

Table 3: Maybe "outcome parameters"

Table 3: Why is the low flow oxygen and HFNC/etc in italic?

Table 4: Bracket missing within the age line.

R: Dear reviewer, we are really grateful for all your efforts to improve our manuscript and its quality. We revised all the manuscript also for English language. However, if accepted the manuscript will be further revised for English language. Thank you very much.

Reviewer 2 Report

The authors researched the clinical outcome of COVID patients in treatment with different steroid therapy care. The study was retrospective and observational conducted in two university hospitals. The steroid therapy used where dexamethasone and methylprednisolone. Main end-point was 30-day mortality.  The study highlighted a better outcome with high-dose methylprednisolone with mortality reduction. This study sheds some light on the treatment strategy for COVID-19 and its outcome implication. The research is easy to read and accessible to a broad class of readers.

The idea of this research is interesting, however I have some minor suggestions to make:

 -The title should be modified including the steroid therapy use

-The protocols of patient admission and management of the hospitals in the study are missing. What were the criteria for treatment with steroid used. What are the antibiotics used in the these patients are also missing. 

-COVID severity and clinical outcome have been associated with Chest CT scan. Can these findings be added and associated with your findings? 

-Limitations of the study are missing

Author Response

The authors researched the clinical outcome of COVID patients in treatment with different steroid therapy care. The study was retrospective and observational conducted in two university hospitals. The steroid therapy used where dexamethasone and methylprednisolone. Main end-point was 30-day mortality.  The study highlighted a better outcome with high-dose methylprednisolone with mortality reduction. This study sheds some light on the treatment strategy for COVID-19 and its outcome implication. The research is easy to read and accessible to a broad class of readers.

The idea of this research is interesting, however I have some minor suggestions to make:

 -The title should be modified including the steroid therapy use

R: Dear reviewer, thank you very much for all your suggestions. We modified the title as required.

-The protocols of patient admission and management of the hospitals in the study are missing. What were the criteria for treatment with steroid used. What are the antibiotics used in the these patients are also missing. 

R: thank you for these suggestions. About management, we reported in Methods that our patients were managed according with Russo A, et al. Comparison Between Hospitalized Patients Affected or Not Affected by Coronavirus Disease 2019. Clin Infect Dis. 2021;72:e1158-e1159. doi: 10.1093/cid/ciaa1745 (now reference n°9). About criteria for steroid therapy, we report now references n°10. About antibiotics, we reported a brief data in Results section (see page 4, lines 129-130).

-COVID severity and clinical outcome have been associated with Chest CT scan. Can these findings be added and associated with your findings? 

R: this is an important observation. Unfortunately, we did not record for all patients the radiological findings, especially chest CT scan. Then, we did not consider these data to reduce a potential bias.

-Limitations of the study are missing

R: thank you again. The limitations are reported in Discussion section (see page 11, lines 216-223) according to the recommendations of Docherty and Smith: BMJ 1999;318:1224-5.

Round 2

Reviewer 2 Report

The authors have addressed the required points. The article is fit for publication.